# Comparison of Overall Sensitivity and Specificity across Different Newborn Screening Algorithms for Congenital Cytomegalovirus

**DOI:** 10.3390/ijns9020033

**Published:** 2023-06-14

**Authors:** Mark R. Schleiss, Lori Panther, Sandeep Basnet, Meklit Workneh, John Diaz-Decaro

**Affiliations:** 1Department of Pediatrics, University of Minnesota Medical School, Minneapolis, MN 55455, USA; 2Moderna, Inc., Cambridge, MA 02139, USA

**Keywords:** congenital CMV, newborn screening for congenital CMV, diagnosis of cCMV, universal screening, targeted screening

## Abstract

Screening newborns for congenital cytomegalovirus (cCMV) infection is critical for early detection and prompt diagnosis of related long-term consequences of infection, such as sensorineural hearing loss and neurodevelopmental delays. The objective of this study was to describe the validity of different newborn cCMV infection screening approaches and compare the expected number of cCMV cases detected across targeted and universal screening algorithms. The overall sensitivity (OSn) of targeted screening algorithms that required failure of auditory brain stem response and transient evoked otoacoustic emissions (TOAE; two-fail serial testing) or TOAE only (one-fail serial testing) before diagnostic CMV testing using saliva and urine PCR tests was 79% and 88%, respectively. The OSn for two-fail serial testing with diagnostic CMV testing using dried blood spot (DBS) was 75%. In contrast, OSn was 90% for universal screening (saliva and urine PCR tests) and 86% for universal screening with DBS testing alone. Overall, specificities were 100% across all algorithms. Universal screening using DBS testing and universal screening using saliva and urine testing can potentially detect 312 and 373 more cCMV cases per 100,000 live births, respectively, than two-fail serial testing. Overall, implementing universal cCMV newborn screening would improve cCMV detection, ultimately leading to better health outcomes.

## 1. Introduction

Cytomegalovirus (CMV), a member of the herpesvirus family, is a common virus that can be transmitted from direct contact with saliva or urine, through sexual contact, from breast milk to nursing infants, and through transplanted organs and blood transfusions [1,2]. In most cases, healthy adults infected with CMV are asymptomatic or have mild symptoms [2]. However, CMV can also be transmitted vertically through the placenta during pregnancy, resulting in congenital CMV (cCMV) infection [3,4]. Symptoms of cCMV, which are not always present at birth, can include rash, jaundice, microcephaly, low birth weight, hepatosplenomegaly, seizures, and retinitis [5]. Globally, 0.5–0.7% of infants are born with cCMV infection and approximately 10% of newborns with a cCMV infection that is symptomatic at birth [6,7,8]. Long-term health conditions, including sensorineural hearing loss (SNHL), visual deficits, and neurodevelopmental delays, develop in 40–60% of symptomatic newborns and in 10–15% of asymptomatic newborns [9,10,11,12]. Notably, cCMV is the most common congenital viral infection and the leading non-genetic cause of SNHL in children [13]. Despite the high prevalence of severe long-term complications, routine screening for cCMV is not standard practice in most parts of the world, resulting in a high proportion of undetected infections [14,15,16].

Screening newborns for cCMV infection is critical for early detection and prompt diagnosis of related long-term consequences of infection [15]. Newborns with moderate to severe cCMV infection can be treated with antiviral medications if administered within 1 month of birth, and non-pharmaceutical interventions can also be implemented when appropriate [15,16]. The standard diagnostic testing for cCMV is PCR testing of saliva, which has a sensitivity and specificity of approximately 93% and 100%, respectively, and is usually followed by confirmatory testing of urine [17]. Dried blood spots (DBS) can also be used for cCMV diagnostic testing; this method has similar specificity (100%) but lower sensitivity (~86%) than saliva PCR testing [17]. Testing must occur within the first 21 days of life to confirm a diagnosis of cCMV [17,18].

Two different approaches of newborn screening for cCMV infection are generally used: targeted screening or universal (or routine) screening. A targeted screening approach involves performing diagnostic testing for cCMV in newborns only following failure of standard hearing evaluations: auditory brainstem response (ABR) and transient evoked otoacoustic emissions (TOAE) tests [19]. Targeted screening typically occurs in neonates prior to hospital discharge. A universal screening approach tests for cCMV in all newborns, regardless of the results from hearing evaluations.

Recommendations for cCMV screening vary across countries and regions [15]. Universal newborn screening for cCMV infection has already been implemented in the Canadian province of Ontario, and legislation has recently been passed for implementation in Saskatchewan and in the US states of Minnesota and Connecticut [20,21,22,23]. Targeted newborn screening has been implemented in several US states as well as other parts of the world, such as Italy’s Tuscany region [24,25]. The objectives of this study are to describe the validity of different newborn cCMV infection screening approaches and to compare the expected number of cCMV cases detected across various targeted and universal screening algorithms.

## 2. Materials and Methods

### 2.1. Assessment of Validity

Sensitivity (Sn) and specificity (Sp) measure test validity and can be expressed as probabilities. Sn of a diagnostic test is the probability of a positive test $\left(T^{+}\right)$ conditional to the presence of a disease or outcome of interest $\left(D^{+}\right)$. Therefore, Sn can be expressed as $P\left(T^{+}|D^{+}\right).$ Sp is the probability of a negative test $\left(T^{-}\right)$ conditional to the absence of a disease or outcome of interest $\left(D^{-}\right)$. Sp can be expressed as $P\left(T^{-}|D^{-}\right)$.

In evaluating screening approaches that incorporate more than one diagnostic test, the validity measures of each test should not be treated as independent but rather as dependent probabilities. For example, consider a diagnostic test A that has a Sn_A_ = 90% and a Sp_A_ = 80%, followed by test B that has the same validity, Sn_B_ = 90% and Sp_B_ = 80%. Both diagnostic test A and B have a 90% probability of a positive test in the presence of disease and an 80% probability of a negative test in the absence of disease, meaning that in the case of both tests, neither will truly identify all disease cases, which results in some false-negative results. If test A is conducted first, and only the positive cases are then screened on test B, the overall probability of identifying all disease cases decreases not only because of the false negatives from test A but also from test B. Similarly, the overall probability of ruling out disease increases because test A and test B are mutually independent. Calculations for overall sensitivity (OSn) and overall specificity (OSp) of hypothetical test A and test B conducted in series (or sequential) are shown in Figure 1. These calculations indicate the dependent probability of test A and test B with respect to Sp and the independent probability with respect to Sn [26]. OSn and OSp were assessed across various hearing and universal cCMV infection newborn screening algorithms.

### 2.2. Screening Algorithms

The targeted screening algorithms evaluated were: (1) two-fail serial testing (ABR and TOAE); (2) one-fail serial testing (TOAE), and (3) two-fail serial testing with DBS diagnostic. The two-fail serial testing algorithm required failure of both ABR and TOAE evaluations prior to the diagnostic testing for CMV, while the one-fail serial testing algorithm only required failure of one hearing evaluation (TOAE). The two-fail serial testing algorithm with DBS included two failed hearing evaluations followed by confirmatory testing with DBS diagnostic PCR. The universal screening algorithms evaluated: (1) Universal screening with saliva PCR followed by confirmation with urine PCR; (2) Universal screening with DBS PCR only. Universal screening does not require a “refer” status on the newborn hearing evaluation prior to screening for and confirmation of cCMV infection; instead, in this setting, all newborns are tested for cCMV.

### 2.3. Identification of Hypothetical cCMV Cases Using Algorithm

We determined OSn and OSp for various cCMV screening approaches to calculate and compare the number of cCMV cases that would potentially be identified in a hypothetical population. The hypothetical population was based on the following assumptions:Population of 100,000 live births;Of 100,000 live births, 100–200 babies are born with SNHL [27,28];Of 100,000 live births, 500 babies are born with a cCMV infection [29];Approximately 10% of babies born with cCMV will have symptoms [8]; andSNHL occurs in approximately 20% of cCMV infections, regardless of type of maternal infection [30].

## 3. Results

### 3.1. Validity Measures

Validity measures of the targeted screening algorithm, including Sn and Sp of each component, as well as the overall validity measures are shown in Figure 2. For hearing evaluations, as previously reported [31,32], the Sn and Sp of ABR were 90% and 80%, respectively; TOAE Sn and Sp were 97% and 100%, respectively. The Sn values of CMV saliva PCR and CMV urine PCR tests were 97% and 93%, respectively, and Sp values were 99% and 100%, respectively. Although the individual diagnostic tests evaluated in this targeted algorithm had high sensitivities independent of the other tests, when applied together in a serial diagnostic workup as part of an algorithm to identify cCMV cases, OSn (as expected) decreased due to an increase in false-negative results (Figure 2 and Figure 3).

This algorithm encompasses two larger components: one to identify hearing loss (ABR or TOAE) and the other to identify cCMV (CMV saliva and urine PCR tests). In a targeted screening approach, the latter builds upon the first. The OSn and OSp for the hearing loss component of this algorithm were 87% and 100%, respectively. Once hearing loss was confirmed and the initial diagnostic test using CMV saliva PCR was performed, the OSn and OSp were 85% and 100%, respectively. Finally, when the overall targeted screening algorithm, including CMV detection using urine PCR, was completed, the OSn and OSp were 79% and 100%, respectively (Figure 2). Each successive test in this algorithm decreased OSn. When taking a universal approach where all newborns are screened for cCMV, the OSn and OSp of universal diagnostic testing with saliva and urine PCR were 90% and 100%, respectively (Table 1). Similarly, across various newborn screening algorithms, the OSn ranged from 75% to 90%; the OSp was 100% across all algorithms.

### 3.2. Predicted Hypothetical cCMV Cases Using Algorithm

The number of cCMV cases (in a hypothetical population of 100,000 live births) identified using the newborn screening algorithm differed depending on the specific algorithm used (Table 1). Using a universal screening approach would identify approximately 3–4 times the number of cCMV cases than would be detected using a targeted screening approach.

## 4. Discussion

Our study demonstrated that universal newborn screening for cCMV would detect more cases than targeted newborn screening approaches. Although such a claim may sound intuitive, we provide support by introducing a direct approach in comparing overall probabilities. The use of a targeted screening approach would result in lower OSn of CMV detection relative to individual tests due to false-negative results with each sequential step. We found that universal screening using DBS diagnostic testing and universal screening using diagnostic saliva and urine testing could potentially detect 312 and 373 more cCMV cases per 100,000 live births, respectively, than two-fail serial testing using ABR and TOAE. Furthermore, because targeted screening approaches are only used in newborns with a “refer” status on the newborn hearing screening that indicate hearing loss may be present, asymptomatic cases of cCMV or those with non-hearing-related symptoms would not be detected.

Conversely, all universal screening methods (i.e., DBS and saliva and urine PCR testing) theoretically detect both symptomatic and asymptomatic cCMV-infected newborns. Detection of asymptomatic cCMV cases is critical as these infants may develop severe complications later in life, such as SNHL, visual deficits, and/or neurodevelopmental sequelae [9,11]. Moreover, one study found that valganciclovir modestly improved SNHL and developmental outcomes in infants ≤30 days of age with symptomatic cCMV infection, demonstrating a benefit of a six-month treatment regimen for both total-ear hearing and the Bayley Scales of Infant and Toddler Development, third edition, which assessed neurologic outcomes [33]. Currently, no studies have investigated the efficacy of antiviral treatments started after 30 days of age, highlighting the importance of early detection of CMV. Although any of the targeted screening algorithms should, in theory, detect the majority of newborns with cCMV who exhibit hearing loss at birth, approximately 43% of newborns with SNHL pass their newborn hearing screen after birth only to develop SNHL later [6]. Additionally, the practicality of implementing a targeted screening approach is challenging from an operational perspective considering that both hearing and CMV testing must occur within the first 21 days after birth to accurately diagnose cCMV [18]. Thus, implementing a universal screening program has the potential to identify more cases of cCMV and to reduce the burden of SNHL among newborns and young children. Overall, we acknowledge that most asymptomatic cCMV cases do not lead to severe complications; however, more evidence is needed to inform clinical practice and monitoring of asymptomatic cCMV newborns.

Acceptance of universal newborn screening for cCMV among parents is high, and progress has recently been made towards incorporating targeted and universal screening programs in the US and several other countries [20,21,22,23,24,25,34,35]. In January 2022, the Minnesota Newborn Screening Advisory Committee approved adding cCMV to the state routine screening panel, making Minnesota the first state in the US to begin implementing universal screening for cCMV [20]. More recently, Connecticut has passed legislation to add CMV testing to their newborn screening panel [23]. Other US states, including Florida, Illinois, Iowa, Kentucky, Maine, New York, Pennsylvania, and Utah, require education of pregnant women and/or have mandated targeted screening programs for cCMV [35]. In Canada, Ontario became the first province to mandate universal screening for cCMV in 2019, and Saskatchewan expanded the program in February 2022 [21,22].

Key barriers in the implementation of universal screening include a lack of consensus on the optimal type of CMV testing (DBS vs. saliva and/or urine), competing priorities, and roles and responsibilities of local hospitals and clinics and state laboratories [6]. PCR testing for cCMV using urine or saliva provides the highest sensitivity and specificity; however, tests can be cumbersome to collect, and specimen collection and processing in a universal screening program would incur significant expense and require significant infrastructure [17]. DBS tests are performed on most newborns in the US, but the CMV viral load in blood is much lower compared with urine or saliva, thus, reducing the sensitivity of DBS testing for cCMV diagnosis [17]. Additionally, DBS collection presents some inherent challenges and requires proper training to ensure that usable samples are collected, especially among neonates where collection of an adequate sample can be challenging [36]. However, with clear guidance at the local and state level, universal screening programs for cCMV can seamlessly be implemented into the Recommended Uniform Screening Panel for newborns [37].

The theoretical nature of this research can be included in the interesting discussion on how cCMV screening is implemented. However, it is important to note that the results presented here should be interpreted as a mathematical exercise, and a head-to-head comparison of targeted versus universal screening would be needed to confirm or negate these findings. Further research and testing of universal screening practices for cCMV is warranted to inform any policy changes.

## 5. Conclusions

Implementing universal cCMV newborn screening via urine and saliva PCR or DBS PCR in clinical practice would improve cCMV detection, ultimately leading to better health outcomes for affected infants. Several US states and other regions worldwide have started implementing both targeted and universal screening programs for cCMV. Further research on the successes and pitfalls of these programs is warranted to inform future public health policies and strategies regarding newborn screening for cCMV.

## Figures and Tables

**Figure 1 IJNS-09-00033-f001:**
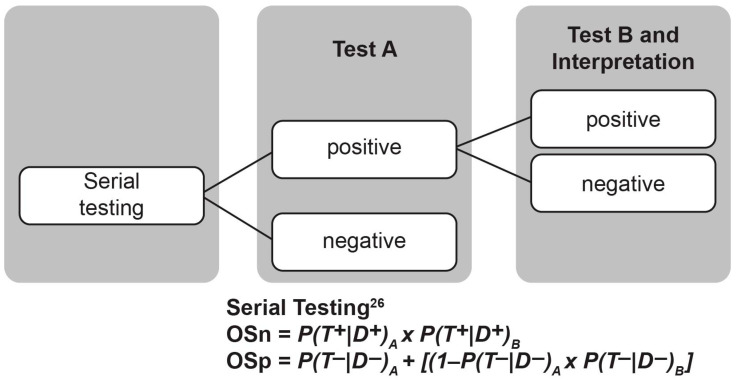
Calculation of OSn and OSp validity measures using serial testing of hypothetical test A and test B. *D*, disease; OSn, overall sensitivity; OSp, overall specificity; *P*, probability; Sn, sensitivity; Sp, specificity; *T*, test.

**Figure 2 IJNS-09-00033-f002:**
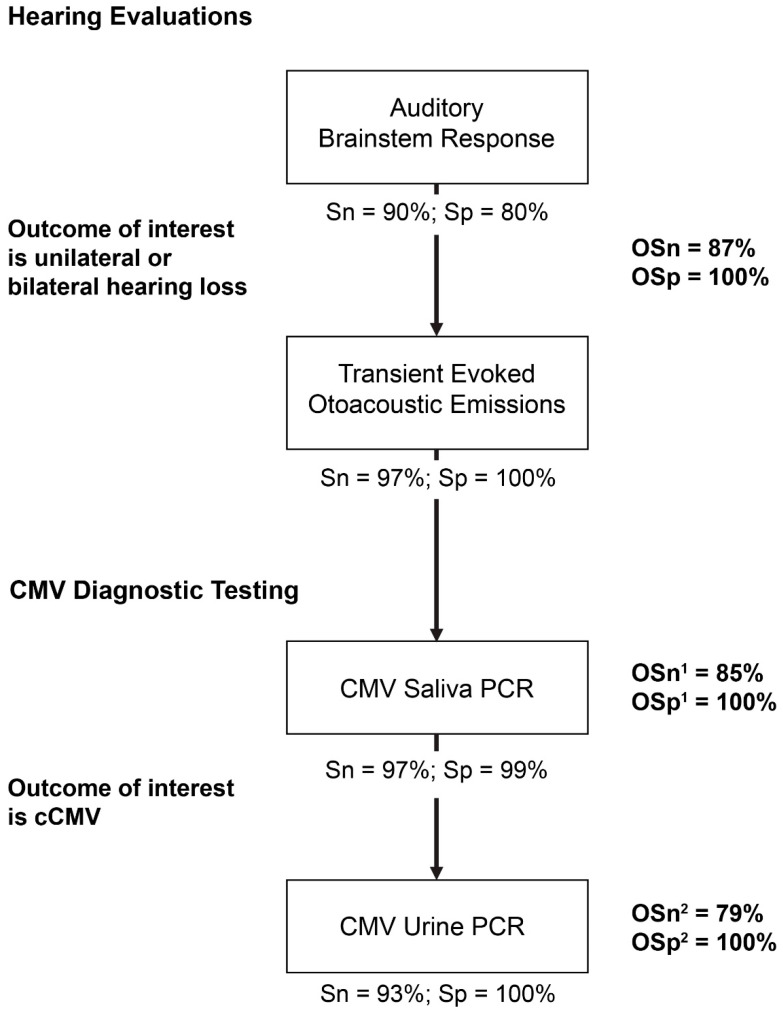
Targeted screening algorithm and associated validity measures. cCMV, congenital CMV; CMV, cytomegalovirus; OSn, overall sensitivity; OSp, overall specificity; PCR, polymerase chain reaction; Sn, sensitivity; Sp, specificity. ^1^ Serial testing of hearing evaluation screening and initial CMV saliva PCR test. ^2^ Serial testing of initial CMV saliva PCR test, followed by confirmatory CMV urine PCR test.

**Figure 3 IJNS-09-00033-f003:**
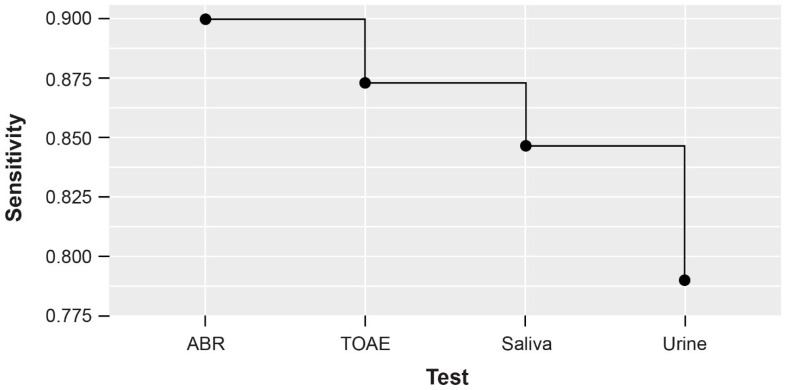
OSn using serial testing with a targeted screening algorithm based on sequential hearing evaluation using ABR and TOAE, followed by CMV screening with saliva PCR test and confirmatory urine PCR test [OSn = *P*(*T*^+^|*D*^+^)_*A*_ × *P*(*T*^+^|*D*^+^)*_B_*]. ABR, auditory brain stem response; TOAE; transient evoked otoacoustic emission.

**Table 1 IJNS-09-00033-t001:** Overall validity measures and identified cCMV cases across various screening algorithm types.

	Targeted Screening Algorithms			Universal Screening Algorithms	
	Two-Fail Serial Testing ^1^	One-Fail Serial Testing ^2^	Two-Fail Serial Testing + DBS ^3^	Universal Screening (Saliva + Urine PCR)	Universal Screening (DBS)
Validity measure					
OSn (%)	78.8	87.6	75.2	90.2	86.0
OSp (%)	100	100	100	100	100
Cases of cCMV infection ^4^					
Cases identified	118	145	111	491	430

cCMV, congenital cytomegalovirus; DBS, dried blood spot; OSn, overall sensitivity; OSp, overall specificity; PCR, polymerase chain reaction. ^1^ Two failed hearing evaluations (auditory brain stem response and transient evoked otoacoustic emissions ((TOAE)). ^2^ Only one failed hearing evaluation (TOAE). ^3^ Two failed hearing evaluations followed by confirmatory testing with DBS diagnostic PCR. ^4^ In a hypothetical population of 100,000 live births, with 150 babies born with hearing loss and 500 babies born with cCMV infection.

## Data Availability

The authors declare that the data supporting the findings of this study are available within this article.

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
