# Peer review of "Comparison of Overall Sensitivity and Specificity across Different Newborn Screening Algorithms for Congenital Cytomegalovirus"

_2409-515X, 2023, doi:10.3390/ijns9020033_

Round 1
Reviewer 1 Report
I enjoyed reading this "estimate" (probability of detecting a congenital CMV (cCMV) baby) of sensitivity and specificity of the various screening strategies for detecting congenital CMV. The methodology would benefit from scrutiny from experts in this area of ‘sensitivity and specificity’ analysis. If valid, the figures are useful for clinicians to review practice. It also provides useful information that can inform policy and allows health care cost estimates by different strategies and types of testing.
As stated, it seems intuitive that Universal Screening is the better (best) strategy + use of urine or saliva samples for CMV PCR having the highest capture rate and yield within this strategy. However, it is useful to have the ‘probabilities’ (that a congenital CMV child will be identified) worked out and presented as such with comparisons made between all the strategies.
The parameters used are ‘known’ assumptions. These assumption are accepted estimates taken from current and relevant literature e.g. the prevalence of hearing loss (1 – 2:1000), birth prevalence (1:200 live births), 10% symptomatic cCMV, SNHL 20% regardless of type of maternal infections. They have used algorithms at entry by hearing screening (targeted or modified target) vs test all (= Universal screening) and then by testing either the dried blood spot (= blood, but dried, spotted, variable, probably low viral loads) or by actual sampling of the baby by urine or saliva.
The authors explain their methodology for calculation of the probability estimates and comparisons of tests.
Some specific comments:
1) In general, the authors explain what ‘targeted screening’ is and what ‘universal screening’ is. However, I feel the “Screening algorithms” could so with some clarity.
The targeted screening is could be better clarified. In general this concept is also poorly clarified in the literature in any case. The use of the word ‘targeted’ in this context just means “targeting the child with hearing loss for CMV testing”. And there are different time possible for this, as outlined by authors, but could be better clarified.
It is my understanding that the standard of care is TOAEs are done first. If the first TOAE fails, then a second is done to confirm the first is a true fail, before an ABR is done. The authors have used the sequence incorrectly (or have I misinterpreted?)
2) Is the “modified” targeted testing something that was made up by the authors? You have cited Reference 1 but it is meaningless and not accessible to the average reader (I certainly don’t know how to access it despite actually trying to look in the StatPearls site). So, only one hearing screen (TOAE) fail? Is “modified targeted testing with DBS” the authors’ terminology (so, modified targeted testing but using DBS as the CMV testing sample/source?
3) The authors should also state that testing (CMV PCR of saliva, then urine) must take place in the first 21 days of life. They allude to timely testing but I don’t believe I see the definition of timely. This is obvious to us in the congenital CMV field but needs stating. Aside from highlighting that is a prerequisite for accurate diagnosis, it serves to show that this is also a level of complexity
4) I find the references Ok but could be improved. E.g. ref 7 for the birth prevalence is a CDC link. There are many more comprehensive birth prevalence or systematic reviews around. This would warrant a better reference
5) How was the algorithm 3 - the ‘stepwise sensitivity comparing hearing screen with sample testing derived? My apologies if this is standard methodology. These sorts of analysis require know how
6) I am glad that the authors have stated that DBS testing has some limitations in sensitivity. It is also a laboratory skill-dependent test
7) There is no discussion about the limitations associated with the authors’ use of this method of analysis. What would you say are limitations or parameters that warrant attention if adopting this type of sensitivity / specificity analysis specific to the congenital CMV cohort. In addition, the ease of trying to capture infants for testing in the first 21 days of is not accounted for in the 'targeted testing' algorithm. You would need to have the ABR done by day 21 AND testing for CMV by day 21 of life (!). in fact - this is the challenge in 'targeted testing' protocols. One study from Australia documented that targeted testing (in their case, testing ONLY after the ABR had identified the hearing loss, only captured ~ 40% of the cohort for timely CMV testing as the babies were > 21 days by the time the ABR was performed (and by definition, CMV testing was also > 21 days and so difficult to prove was congenitally acquired if present) - see reference below. I feel this issue of timely testing (by 21 days) and fitting in with your algorithms in 'targeted testing' has been largely ignored in your discussion. I am not sure if it needed to be factored into the Methodology.
Rawlinson, W.D, Palasanthiran P, Hall B et al., et al. Neonates with congenital Cytomegalovirus and hearing loss identified via the universal newborn hearing screening program. Journal of Clinical Virology 102, 110-115 (2017).
Reviewer 2 Report
Summary
This manuscript describes a theoretical comparison of different newborn screening (NBS) algorithms used to detect congenital cytomegalovirus (cCMV) infection. A comparison is made among three choices using first tier newborn hearing screening and two using first tier laboratory tests. Using published data on the sensitivity and specificity of each test alone, the authors compute an overall sensitivity and specificity for each algorithm. They present the theoretical results of screening 100,000 newborns.
Strengths
Universal cCMV testing is being considered by many jurisdictions and a rational approach to choosing the best algorithm is important. Other than important concerns stated below about the terminology the authors have chosen, the manuscript is well written and appropriately brief.
Areas for Improvement
The title makes clear that the paper is about NBS for cCMV, but the manuscript gets lost in its choice of terminology. For example, “targeted hearing screening” is a misnomer. Universal hearing screening is often followed by targeted cCMV screening. Overall, screening for cCMV can be done as a targeted screen (those who fail newborn hearing screening) or universally. Changing the wording for the five approaches would not change the associated models and calculations. The headings in Table 1 should be “Targeted Screening Algorithms” and “Universal Screening Algorithms”. “Targeted Hearing” should be “First Tier Hearing”, etc. Making these revisions changes much of section 2.2.
The advantage of universal cCMV NBS over targeted CMV testing is clearly shown in the bottom row of Table 1 as being about a 200% increase in cases detected. Line 177 of the Discussion states that the gains are “7.2% and 11.4%”, which would probably not be sufficient to warrant the increased cost of universal cCMV screening. I don’t understand the discrepancy.
Minor Points
I do not think figure 2 contributes to the manuscript because it only displays one of the five algorithms. If a figure could be devised to display all five, that would be helpful.
Likewise, I don’t think figure 2 adds anything to the description in the text.
I think any report focusing on evaluating tests and algorithms needs to make a statement about the “gold standard” (sensitivity and specificity 100%) if one is known or the accepted best measure if there is no “gold standard”. I think the “gold standard” for cCMV is urine PCR. References are given for the sensitivity and specificity of the hearing tests, but I could not find references for the lab tests (DBS, saliva and urine PCR).
The numerical expressions should be made consistent. For example, bullet 2 (line 111) should say, “Of 100,000 live births, 100-200 are born with SNHL”. Bullet 3 should say, “Of 100,000 live births 500 are born with cCMV.” Making these statements consistent removes the need for the last paragraph of section 2, lines 115-117. Likewise, sensitivities and specificities should be expressed either as percentages or decimal fractions (Figure 2) but not both.
DBS CMV PCR is probably not considered a confirmatory test despite its reported 100% specificity in some labs. Urine PCR is probably the only confirmatory diagnostic test so far.
Everything after the first sentence of section 3.2 (lines 157-161) is just repeating the data shown nicely in Table 1 and can be deleted.
The “sentence” that starts, “Moreover” on line 186 is not a sentence.
Reviewer 3 Report
This is an exceptionally well written manuscript which was a pleasure to read. The manuscript is timely as cCMV screening is a very topical at present.
My only comments are:
1. It would be useful to include additional detail on the efficacy of treatment of cCMV with antivirals to put the case for screening in context. On page 6, lines 186 to 188 there is mention of treatment but the sentence doesn't make sense and would benefit from some additional detail.
2. In countries/states where screening for cCMV is not in place, it is not uncommon for the NBS lab to receive a request to release the NBS sample for CMV testing. In some labs (in the UK anyway) this has become quite a time consuming activity as the number of retrospective requests has increased and the majority of samples need to be retrieved from archives. This may be worth commenting on in the discussion.
Reviewer 4 Report
This study is of important value since it demonstrates that implementing universal CMV newborn screening would improve CMV detection and health outcomes.
Round 2
Reviewer 2 Report
Summary
This manuscript describes a theoretical comparison of different newborn screening (NBS) algorithms used to detect congenital cytomegalovirus (cCMV) infection. A comparison is made among three choices using first tier newborn hearing screening and two using first tier laboratory tests. Using published data on the sensitivity and specificity of each test alone, the authors compute an overall sensitivity and specificity for each algorithm. They present the theoretical results of screening 100,000 newborns.
Strengths
Universal cCMV testing is being considered by many jurisdictions and a rational approach to choosing the best algorithm is important. Most of the manuscript is well written and appropriately brief.
Areas for Improvement
The phrase “targeted hearing” has been corrected in some, but not all places in the manuscript. It misstates what is done because the hearing screening being discussed is always universal and its results are used to target the cCMV screening.
Minor Points
The confusing 7.2% and 11.4% were made clearer in the Discussion, but are still in the Abstract and should be changed to match the Discussion.
I think “and confirmation” is redundant in line 103. In newborn screening, “diagnostic testing” refers to whatever tests are done to confirm or exclude the suspected diagnosis.
I don’t understand “hearing loss” in the last sentence of the first paragraph of the Discussion, line180.
The Discussion appropriately includes possible visual deficits in the complications of cCMV and that should also be added to the list in the Introduction line 39.
Author Response
Reviewer 1:
Summary
This manuscript describes a theoretical comparison of different newborn screening (NBS) algorithms used to detect congenital cytomegalovirus (cCMV) infection. A comparison is made among three choices using first tier newborn hearing screening and two using first tier laboratory tests. Using published data on the sensitivity and specificity of each test alone, the authors compute an overall sensitivity and specificity for each algorithm. They present the theoretical results of screening 100,000 newborns.
Strengths
Universal cCMV testing is being considered by many jurisdictions and a rational approach to choosing the best algorithm is important. Most of the manuscript is well written and appropriately brief.
Areas for Improvement
The phrase “targeted hearing” has been corrected in some, but not all places in the manuscript. It misstates what is done because the hearing screening being discussed is always universal and its results are used to target the cCMV screening.
Response: Thank you for highlighting this discrepancy. We have modified the text throughout the manuscript to replace “targeted hearing” with “targeted screening.”
Minor Points
The confusing 7.2% and 11.4% were made clearer in the Discussion, but are still in the Abstract and should be changed to match the Discussion.
Response: We have updated the text within the abstract on page 1, lines 20-22 to align with the discussion: “Universal screening using DBS testing and universal screening using saliva and urine testing would potentially detect 312 and 373 more cCMV cases per 100,000 live births, respectively, than 2-fail serial testing.”
I think “and confirmation” is redundant in line 103. In newborn screening, “diagnostic testing” refers to whatever tests are done to confirm or exclude the suspected diagnosis.
Response: We have removed “and confirmation” from the statement on page 3, lines 116-119. The sentence now reads: “The 2-fail serial testing algorithm required failure of both ABR and TOAE evaluations prior to diagnostic testing for CMV, while the 1-fail serial testing algorithm only required failure of one hearing evaluation (TOAE).”
I don’t understand “hearing loss” in the last sentence of the first paragraph of the Discussion, line180.
Response: We have revised the statement on page 7, lines 194-197 for clarity: “Furthermore, as targeted screening approaches are only used in newborns with a “refer” status on the newborn hearing screening that indicate hearing loss may be present, asymptomatic cases of cCMV, or those with non−hearing-related symptoms, would not be detected.”
The Discussion appropriately includes possible visual deficits in the complications of cCMV and that should also be added to the list in the Introduction line 39.
Response: We have updated the statement on page 1, lines 38-40 to read: “Long-term health conditions, including sensorineural hearing loss (SNHL), visual deficits, and neurodevelopmental delays develop in 40%–60% of symptomatic newborns and in 10%–15% of asymptomatic newborns [9-12].”
